# Higher order effects in organic LEDs with sub-bandgap turn-on

Sebastian Engmann [1,2], Adam J. Barito[2], Emily G. Bittle [2], Noel C. Giebink [3], Lee J. Richter[4] & David J. Gundlach[2]

Spin-dependent nonlinear processes in organic materials such as singlet-fission and triplet-triplet annihilation could increase the performance for photovoltaics, detectors, and light emitting diodes. Rubrene/$C_{60}$ light emitting diodes exhibit a distinct low voltage (half-bandgap) threshold for emission. Two origins for the low voltage turn-on have been proposed: (i) Auger assisted energy up-conversion, and (ii) triplet-triplet annihilation. We test these proposals by systematically altering the rubrene/$C_{60}$ interface kinetics by introducing thin interlayers. Quantitative analysis of the unmodified rubrene/$C_{60}$ device suggests that higher order processes can be ruled out as the origin of the sub-bandgap turn-on. Rather, band-to-band recombination is the most likely radiative recombination process. However, insertion of a bathocuproine layer yields a 3-fold increase in luminance compared to the unmodified device. This indicates that suppression of parasitic interface processes by judicious modification of the interface allows a triplet-triplet annihilation channel to be observed.

[1] Theiss Research, 7411 Eads Avenue, La Jolla, CA 92037, USA. [2] Nanoscale Device Characterization Division, National Institute of Standards and Technology, 101 Bureau Drive, Gaithersburg, MD 20899, USA. [3] Department of Electrical Engineering, The Pennsylvania State University, Electrical Engineering West, State College, PA 16801, USA. [4] Materials Science and Engineering Division, National Institute of Standards and Technology, 101 Bureau Drive, Gaithersburg, MD 20899, USA. Correspondence and requests for materials should be addressed to S.E. (email: sebastian.engmann@theissresearch.org) or to D.J.G. (email: david.gundlach@nist.gov)

Spin-transport in organic materials and coherent spin processes like singlet-fission (SF) and triplet-triplet annihilation (TTA) have gained recent interest in the scientific community due to the possibility to observe and study quantum phenomena at room temperature. Additionally, the comparably long relaxation times in organic materials may enable quantum-based information storage and logic devices. In the established areas of organic photovoltaics (OPVs) and organic light-emitting diodes (OLED), SF and TTA are promising mechanisms for enhanced device performance. SF enables increased current density in photovoltaic applications and high quantum yields and increased response in detector applications. In OLEDs, TTA enables device performance above the 25% emission efficiency limit of singlet emitters, as three-fourth of injected carriers form triplets and only ¼th will form singlet exciton states. Another benefit of triplet-based OLEDs is a possible reduction in the drive voltage that, despite recent success of OLEDs in small format displays, remains relatively high compared to LEDs based on compound semiconductors like GaAs or InGaAs. In OLEDs, the applied potential for effective emission is often observed to be on the order of the optical bandgap[1–3] that in turn leads to confusion and potential misinterpretation if a "sub-bandgap" or low turn-on voltage is observed.

Recently, OLED devices have been reported with electro-luminescence (EL) turn-on at voltages less than half of the optical gap based on both polymer[4] and small-molecule emitters[5]. One of the most commonly studied devices uses the small-molecule rubrene as emitter and hole transporter and the fullerene $C_{60}$ as the electron transporter. In these devices, two distinct mechanisms have been proposed for the low-voltage EL: an Auger-assisted energy up-conversion process at the heterojunction interface[6,7], or Dexter transfer of triplet charge transfer (CT) states into triplet exciton states, followed by TTA to produce an emitting singlet[8–10]. In both cases, the non-linear charge dynamics at the rubrene/$C_{60}$ interface are crucial to the mechanism.

In this report, we study OLED devices where we systematically alter the rubrene/$C_{60}$ interface to vary the CT-state formation and recombination rates via the introduction of (i) a thin bath-ocuproine (BCP) interlayer (1–5 nm), that should significantly suppress CT-state formation from singlets and CT-state non-radiative recombination or (ii) a mixed rubrene/$C_{60}$ interlayer that should greatly enhance the role of the interface in total device performance. Our results show that non-linear processes such as TTA and Auger are inconsistent with the device current–voltage characteristics. However, when parasitic interface pathways are suppressed, evidence for a TTA process is observed.

## Results

**Qualitative device measurements.** Heterojunction OLEDs based on rubrene/$C_{60}$ interfaces with or without a thin (0–5 nm) inter-layer of BCP or co-evaporated rubrene:$C_{60}$ (1:1 by mass ratio) was fabricated. The active layers were evaporated onto Glass/ITO/PEDOT:PSS:Nafion substrates, with PEDOT:PSS:Nafion acting as the hole injection layer (HIL), and capped by BCP/Al acting as electron injection layer. The composition of the mixed PEDOT:PSS:Nafion HIL was tuned to provide a higher work function (−5.3 eV to −5.8 eV[11–15]) and closer energetic match to the highest occupied molecular orbital, HOMO, of the rubrene emission layer. Throughout the manuscript we will use the chemical nomenclature of HOMO to describe the ionization potential and hole transport level and lowest unoccupied molecular orbital (LUMO) to describe the electron affinity and electron transport level. Bandgap will refer to the HOMO–LUMO separation, while optical gap will refer to the absorption onset. Shown in Fig. 1 are the chemical structures of rubrene and $C_{60}$ as well as the architectures used for the devices reported within this manuscript.

Shown in Fig. 2 are the current density, $J(V)$-characteristics, and luminance, $L(V)$-characteristics, for control devices fabricated with no $C_{60}$ ETL (rubrene only) and with no interface modification (BCP, 0 nm). Also shown are the BCP interlayer and

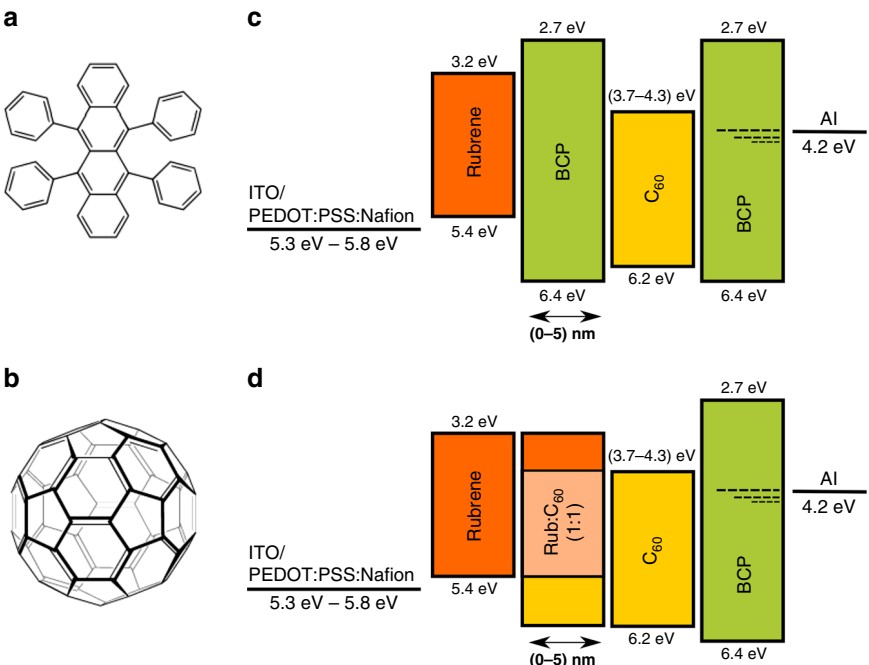

**Fig. 1** Molecular structures and energy level diagrams. Molecular structures of (**a**) rubrene and (**b**) $C_{60}$. Energy level diagrams of devices with (**c**) BCP interlayer and (**d**) mixed rubrene:C60 (1:1) interlayer. Highest occupied molecular orbital (HOMO) and lowest unoccupied molecular orbital (LUMO) values for rubrene and $C_{60}$ taken from refs. [6, 7, 10, 51], BCP values taken from refs. [18, 19]. Range of commonly reported work functions for PEDOT:PSS:Nafion, for 10% Nafion, the expected work function is near 5.5 eV[11–15]

rubrene:$C_{60}$ interlayer devices. The reference rubrene-only device exhibits a forward bias turn-on of the injection current near 2.0 V and of the luminescence near 2.8 V, which is above the ≈2.2 eV optical bandgap of rubrene. The observed $L(V)$-characteristics match the results by He et al.[16]. The rubrene/$C_{60}$ control device exhibits a forward bias turn-on near 0.8 eV, and an earlier luminescence turn-on voltage of about 1 V, coinciding with half the optical gap of rubrene, in close agreement with earlier literature reports[6–10]. Similarly, the variation and magnitude of the magneto-electroluminescence (MEL) response (Supplementary Note 1 - Supplementary Figure 3) is in accord with earlier literature[9]. The introduction of the thin BCP interlayers or thin rubrene:$C_{60}$ intermixed layers does not significantly alter the $J(V)$-characteristics of the devices; however, a significant effect on

the luminescence of the OLED is observed. The mixed interlayer suppresses luminance by an order of magnitude, while an increase in emitted light intensity can be observed for thin (1–3) nm BCP interlayers. In contrast, a thick >5 nm BCP interlayer decreases the device luminance.

For all devices studied, the spectral response is invariant and matches the reported EL and photoluminescence spectra for rubrene (Fig. 3 and Supplementary Figure 2)[5,6]. In an earlier study of thick (40 nm) mixed interlayers, strong emission at ~870 nm was observed and attributed to CT-state emission[17]. We observe a very weak luminescence feature at 870 nm (≈1.4 eV) in the case of the 5 nm thick rubrene:$C_{60}$ mixed interlayer. The very similar $J(V)$-characteristics of the devices with and without BCP interlayers (<5 nm) suggests that charge carrier transport across the nominally insulating layer is possible and is only compromised for thick BCP layers. This may be due to a trap assisted hopping of electrons across the interlayer similar to that observed at the $C_{60}$/BCP/metal electrode[18,19], direct tunneling, or percolation across a non-continuous layer. In devices limited by interface recombination[20], an exponential decrease in current with spacer thickness has been reported.

The devices in Fig. 2 were additionally characterized with respect to their performance as heterojunction solar cells. Shown in Fig. 4 are the $J(V)$-characteristics under AM 1.5 illumination. With increasing BCP interlayer thickness, a monotonic decrease in short-circuit current density ($J_{sc}$) is observed. The devices with mixed rubrene:$C_{60}$ interlayers (2 and 5) nm exhibit similar photocurrent relative to the rubrene/$C_{60}$ control, suggesting that absorption and/or exciton transport in the rubrene is current limiting.

Shown in Fig. 5 is a summary of the OLED- and OPV-relevant characteristics for the various BCP and rubrene:$C_{60}$ interlayer thicknesses. The luminescence at 2 V bias shows a maximum at about 2–3 nm of BCP interlayer thickness, corresponding to a threefold increase compared to the reference device. In contrast, the introduction of the mixed rubrene:$C_{60}$ layer leads to a radical decrease in luminescence with interlayer thickness. This is in agreement with recent observations of Chen et al.[9] for a mixed rubrene:$C_{60}$ active layer. Strikingly, the short-circuit density in PV operation of the mixed interlayers seems to be independent of

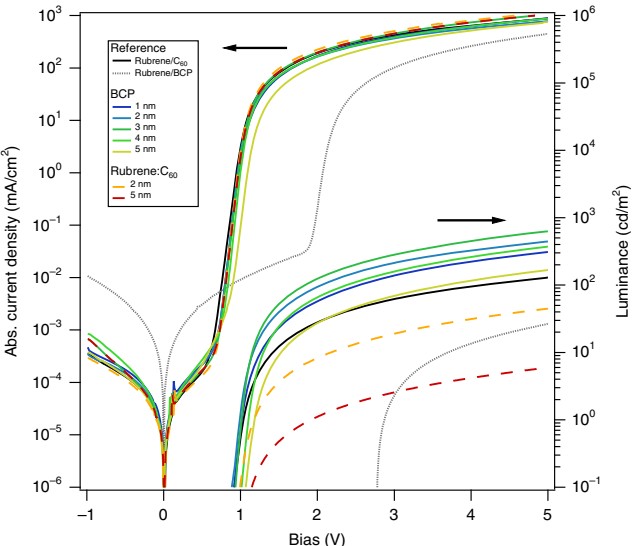

**Fig. 2** OLED device characteristic. $J(V)$- and $L(V)$-characteristics of rubrene/$C_{60}$ OLEDs with BCP or rubrene:C60 (1:1) interlayers inserted between the rubrene and $C_{60}$ layers. A plot on linear scale can be found in Supplementary Figure 1

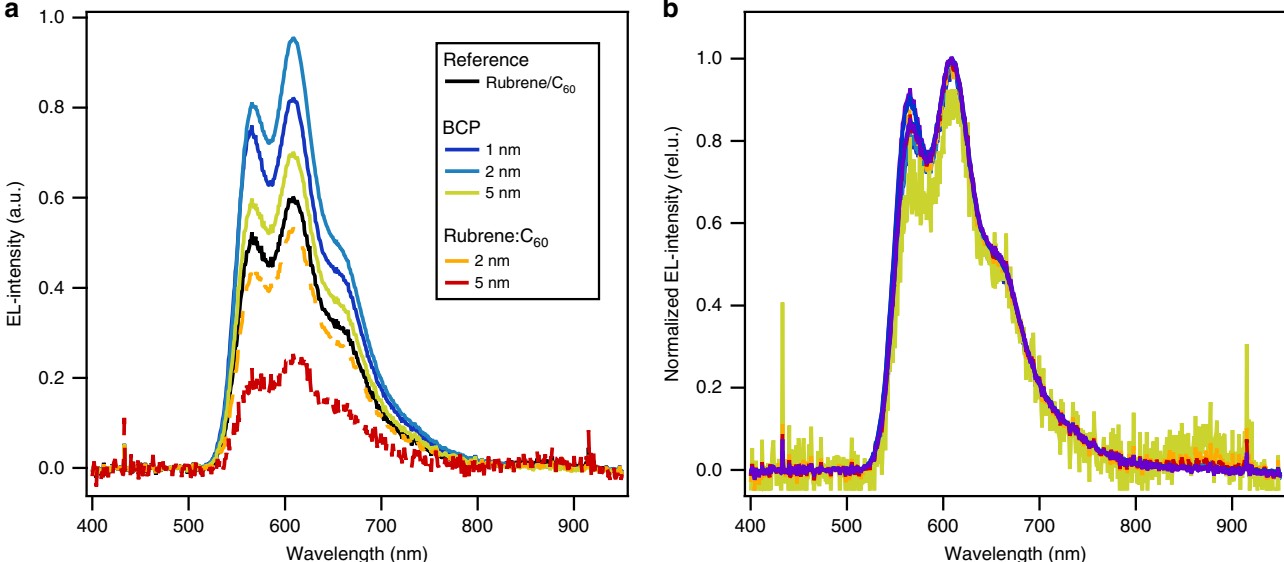

**Fig. 3** Emission spectra. **a** Electroluminescence spectra of rubrene/$C_{60}$ OLEDs with BCP or rubrene:C60 (1:1) interlayers inserted between the rubrene and $C_{60}$ layers. For comparison, a control device (rubrene/$C_{60}$) with no interlayer is shown. **b** The emission data normalized to the maximum (in the wavelength range 600–610 nm). All spectra were recorded at a current level of 18.75 mA/cm$^2$

interlayer thickness, but decreases exponentially with BCP interlayer thickness. CT states are bound by long-range Coulombic interactions, thus the influence of a spacer can be complex, modifying both CT-state binding energy[21] and formation and dissociation rates[22,23]. As will be discussed later, the exponential decrease in OPV $J_{sc}$ with BCP layer thickness indicates that CT-state recombination is not rate limiting and the increase in luminesce arises from a decrease in parasitic singlet quenching via CT-state formation.

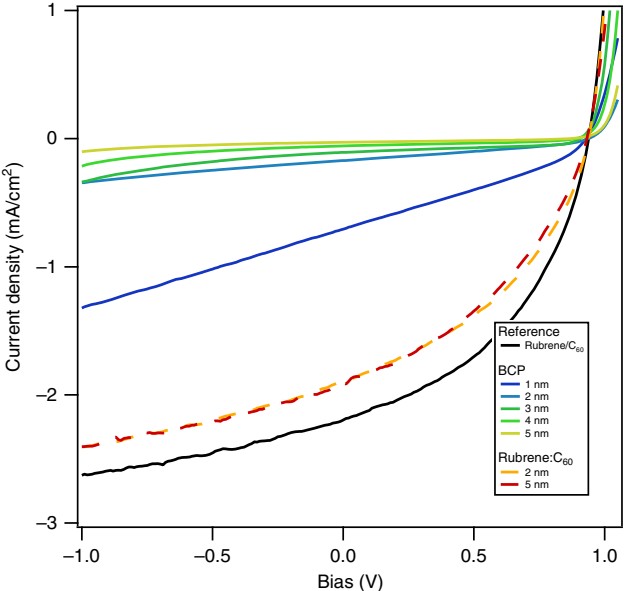

**Fig. 4** OPV device characteristic. $J(V)$-characteristics of rubrene/$C_{60}$ devices operated as solar cells with BCP or rubrene:$C_{60}$ interlayer thicknesses between 1 and 5 nm under AM1.5 conditions. For comparison, a reference heterojunction rubrene/$C_{60}$ without the either of the two interlayers is shown

**A closer look at the sub-bandgap "turn-on".** In order to understand the origin of the sub-bandgap turn-on and further explore the role of higher order recombination processes such as Auger-assisted up-conversion at the heterojunction interface or TTA, we examine the $L(J)$-characteristics and $J(V)$-characteristics more closely. First, we investigate the $J(V)$-characteristics to gain information on the recombination order via analysis of the diode ideality factor $n$. For a classic semiconductor, the diode ideality factor depends on the order of recombination as $n = 2/order$, e.g., $n = 2$ (order $= 1$) for monomolecular (presumably from trap-assisted recombination or surface recombination), $n = 1$ (order $= 2$) for bimolecular, and $n = 2/3$ (order $= 3$) for trimolecular or Auger recombination[24,25]. Although the diode equation and ideality were derived for inorganic pn-junctions with well-defined band structure and delocalized free charge carriers, Giebink et al. have shown that a similar diode equation can be derived for organic heterojunctions where charge transport in an intrinsic material characterized by a transport level is dominated by hopping through tightly bound localized states[26,27]. Again, just like in the inorganic case, bimolecular recombination at the donor/acceptor interface is characterized by an ideality of unity, and trap-assisted recombination yields an ideality larger than unity.

Shown in Fig. 6 are the $J(V)$- and $L(V)$-characteristics of the reference rubrene/BCP, the rubrene/$C_{60}$ heterojunction, and the optimized device with 3 nm BCP interlayer, as well as fits to the $J(V)$ data using a one-diode-equivalent circuit model based on Maxwell–Boltzmann statistics:

$$J = J_0 \left( e^{\frac{V - R_S J}{n V_t}} - 1 \right) + \frac{V - R_S J}{R_p}, \qquad (1)$$

where $R_s$ and $R_p$ are a series and parallel shunt resistance that depend on the morphology within the device as well as macroscopic connections.

The $J(V)$-data can be reasonably well represented by the 1-diode model over a wide range of bias voltages; the results of the fit to the model are summarized in Table 1. We also determine an ideality factor from the $L(V)$ data by fitting the region around the turn-on voltage (1 V for the rubrene/$C_{60}$-based devices and 2 V

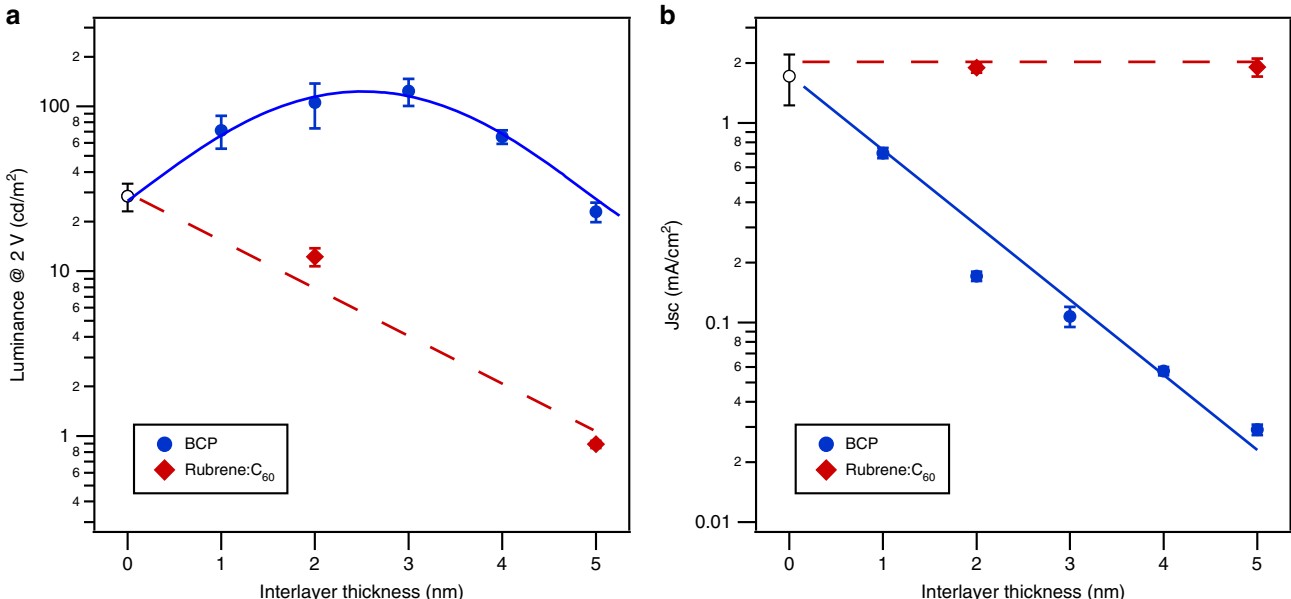

**Fig. 5** Device summary. **a** Luminance of rubrene/$C_{60}$ OLEDs as a function of interlayer thickness. **b** Short-circuit current of rubrene/$C_{60}$ PV devices under AM 1.5 conditions as a function of interlayer thickness. Error bars correspond to the standard deviation of four devices. Lines are guides to the eye

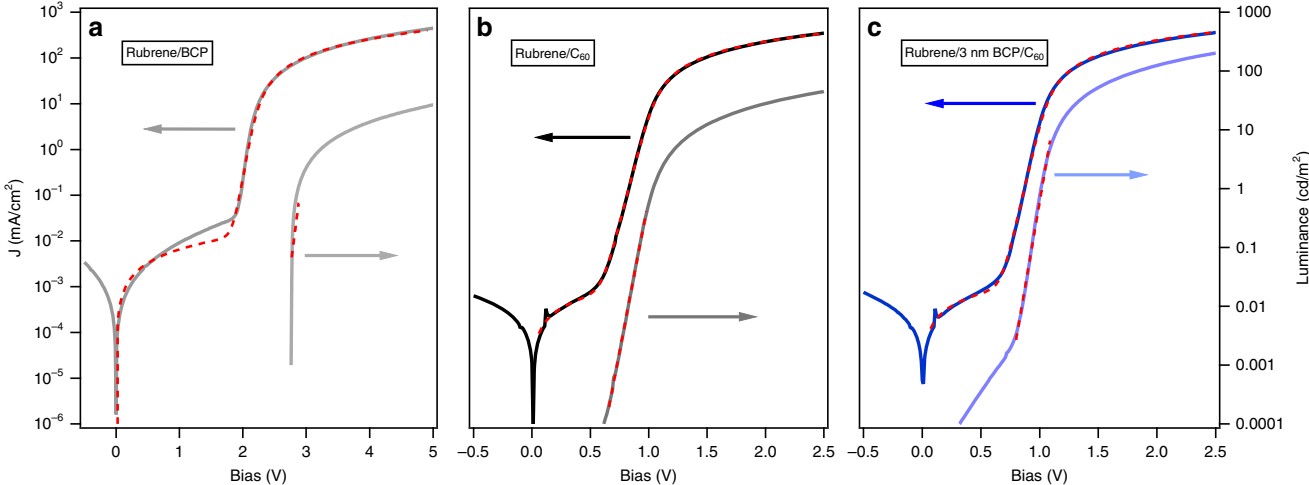

**Fig. 6** Equivalent circuit fit. $J(V)$- and $L(J)$-characteristics of rubrene/$C_{60}$ OLEDs, (**a**) reference device with no C60 ETL, (**b**) baseline device without BCP interlayer, (**c**) optimized device with 3 nm BCP interlayer. Broken red lines correspond to a fit to a 1-diode-equivalent circuit model and modified exponential term in case of the $J(V)$- and $L(V)$-characteristics, respectively

**Table 1 Summary of the 1-diode-equivalent circuit model**

| | $R_s$ (Ω) | $R_p$ (MΩ) | $J_0$ (fA/cm²) | $n_J$ | $n_{Lum}$ |
|---|---|---|---|---|---|
| Rubrene/BCP | 158 ± 13 | 3.8 ± 0.3 | $(1.8 ± 1) × 10^{-5}$ | 2.09 ± 0.03 | 2 ± 0.16 |
| Rubrene/$C_{60}$ | 109 ± 8 | 97 ± 14 | 53 ± 34 | 1.55 ± 0.08 | 1.62 ± 0.13 |
| Rubrene/BCP(3 nm)/$C_{60}$ | 99 ± 6 | 85 ± 12 | 1.3 ± 0.8 | 1.38 ± 0.07 | 1.44 ± 0.11 |

Errors correspond to 1 standard deviation of the fit parameter. The active area of the device is 0.04 cm². The ideality factor $n_{Lum}$ was obtained from a fit to the $L(V)$-characteristics near the turn-on voltage

for the rubrene/BCP device) with an exponential of the form: $L = L_0 \left( e^{\frac{V}{nV_t}} - 1 \right)$. We restrict the range, as for large bias voltages the current density through the device is limited by the series resistance $R_s$, and for small voltages, the current is dominated by $R_p$. Note that the $L(V)$-data after turn-on matches the shape of the $J(V)$ data. The luminance ideality factor is also given in Table 1.

We observe an ideality factor of 2 in case of the rubrene/BCP device, pointing towards trap-assisted recombination or injection-limited current such that the device is limited by monomolecular recombination. Charge carrier injection limitations are supported by the relatively large series resistance. For the two $C_{60}$-based devices, that exhibit half-bandgap thresholds, the diode ideality factor is between 1.4 and 1.6. This effectively rules out Auger recombination and a coherent triplet formation process (see Supplementary Note 2: Considerations on Diode Ideality) as dominant processes in the device at these bias voltages and suggests a combination of mono- and bimolecular recombination as is typically observed in organic devices[28,29].

**A drift-diffusion model without higher order recombination**. The absence of evidence for Auger recombination (diode ideality factor of 2/3) or fourth-order processes (coherent TTA) suggests that higher order up-conversion processes are not required to explain the low-voltage EL and indicates that it can be understood in the same framework as inorganic pn-junction LEDs, where low-voltage EL is commonplace. In a classic pn-junction LED, the EL stems from the ideal diode minority carrier diffusion current (that is simply band-to-band recombination in the quasi-neutral regions, following Eq. (1)) and thus can be observed at arbitrarily low bias set by the sensitivity of the photon detection system. This

is rigorously established through the generalized Planck equation for luminescent radiation derived by Würfel[30] that has been successfully applied to OPV devices[31,32] and is such a common occurrence for inorganic LEDs that it is rarely commented upon[33–37]. Contrary to this, in the field of organic electronics, the built-in potential (built-in voltage $V_{bi}$) is often viewed as difference between the transport levels and as such approximately the bandgap of the semiconductor. This leads to the believe that the turn-on voltage in OLEDs, which is roughly the built-in voltage, must be equivalent to the bandgap of the material. However, from the point of view of a classical homojunction and its adaption to organic materials, the built-in potential is smaller than the bandgap as does the turn-on of the device, see Supplementary Note 3 and Supplementary Figure 4.

To determine if the low-voltage turn-on and half-gap luminesce are consistent with classic heterojunction diodes with reduced built-in voltage due to transport gap offsets, we performed drift-diffusion simulations of the entire device structure using the open source device simulator GPVDM (Ver. 4.98.011) developed by R.C.I. MacKenzie[38–40]. Detailed information on how the simulator solves the drift-diffusion and Poisson-equation can be found elsewhere[41]. For simplicity, we simulate the $J(V)$-characteristics of the reference rubrene/$C_{60}$ device allowing bimolecular recombination, $R = k(np - n_i^2)$, from free holes and electrons only. The bimolecular recombination rate $k$ was assumed to be of the value of Langevin recombination rate determined by the electron and hole mobilities throughout the device. It is well known that organic materials exhibit tail states extending from the transport edge into the bandgap. To effectively simulate transport via these states, we replaced the HOMO/LUMO energies of rubrene and $C_{60}$ with an effective energy representing the tail states. It is reasonable to assume that

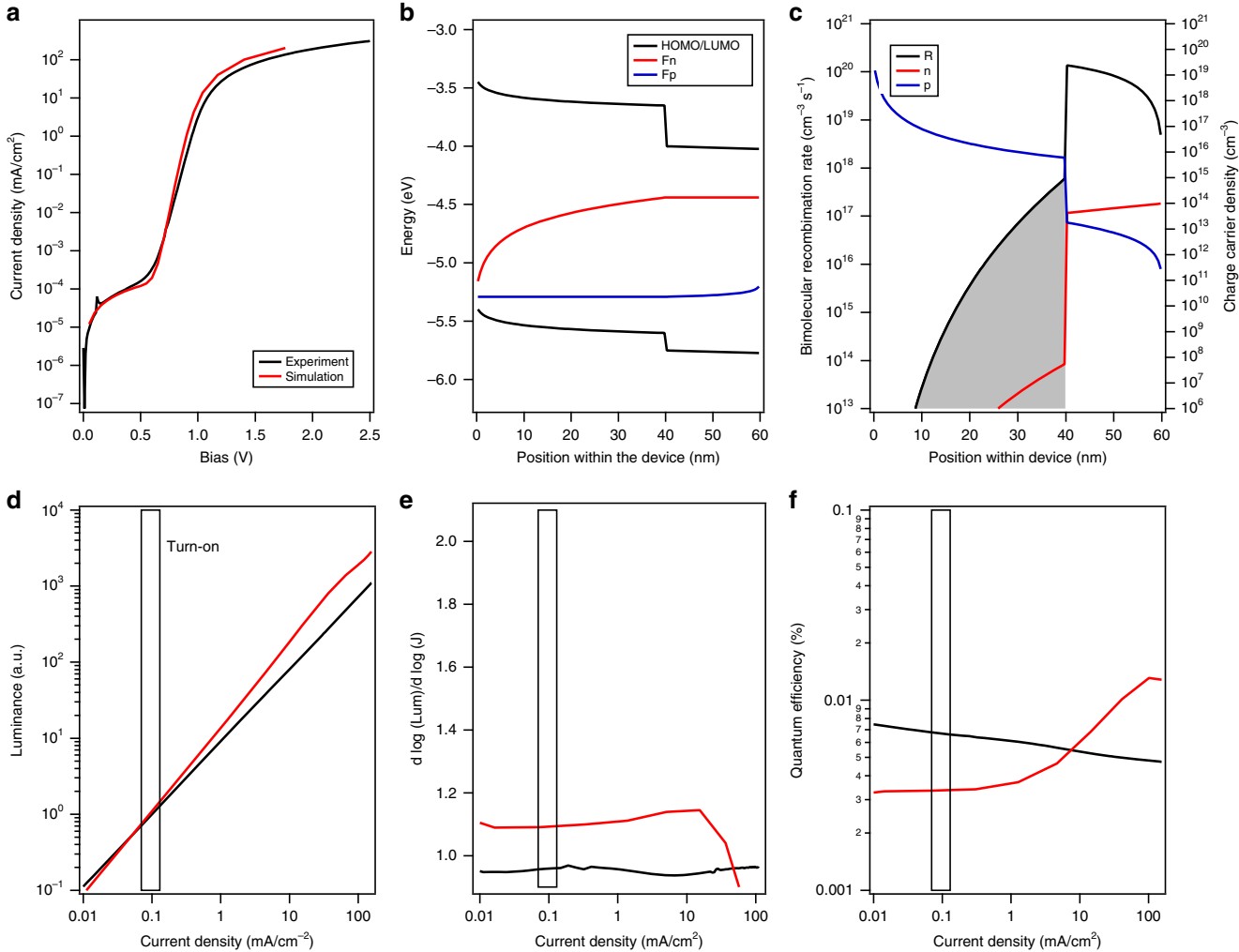

**Fig. 7** Drift diffusion modeling. Simulated and experimental $J(V)$-characteristics of a rubrene/C60 ( 40 nm/20 nm) heterojunction (**a**). Also shown are the band diagram (**b**) and the bimolecular recombination rate within the device (**c**) at 0.8 V. $L(J)$, $\partial \log(L)/\partial \log(J)$, and the internal quantum efficiency are shown in **d**–**f**. In case of the simulation, the internal quantum efficiency, and in case of the experimental data, the external quantum efficiency are shown. Note all recombination within the device is bimolecular recombination of free charge carriers. The simulation included a series resistance of 25 Ω corresponding to the ITO and wire resistances and a 35 MΩ shunt resistance

disorder will lead to $10^{15}\,\mathrm{cm}^{-3}$ to $10^{16}\,\mathrm{cm}^{-3}$ trap states that might reach up to 50–150 meV further into the gap. This leads us to an effective transport gap of 2 eV in case of the rubrene and 1.85 eV in case of $C_{60}$. A detailed summary of all simulation parameters in the GPVDM notation can be found in Supplementary Note 4 - Supplementary Table 1 and Supplementary Table 2.

Shown in Fig. 7 are the simulated $J(V)$-characteristics in direct comparison with the experimental $J(V)$ curve obtained for the rubrene/$C_{60}$ heterojunction device, as well as the obtained luminance and its derived parameters. Note the good agreement between the experimental and simulated diode characteristics. The simulation produces the early-voltage turn-on despite not including any TTA or Auger recombination. Additionally, the model predicts the nominal linear dependence of luminescence on current. However, the diode ideality is not captured perfectly as bimolecular recombination inherently is characterized by an ideality of 1.

**A rate equation-based model.** While neither TTA nor Auger are necessary to describe the current and luminescence of the baseline rubrene/$C_{60}$ device, the increase in luminescence with BCP layer thickness suggests that interface kinetics are relevant to the

interlayer modified devices. We investigate this with a detailed kinetic model for CT-state-mediated singlet production. Assuming the injected current leads to the formation of CT states at the interface and triplets ($T$) and singlets ($S$) in the rubrene layer of the heterojunction device via incoherent pooling, the steady-state rate equations are:

$$\frac{\mathrm{d}n}{\mathrm{d}t} = \frac{\mathrm{d}p}{\mathrm{d}t} = \frac{J}{qa_0} - knp + k_{\mathrm{CT,diss}}\mathrm{CT} = 0, \qquad (2)$$

$$\frac{\mathrm{dCT}}{\mathrm{d}t} = c_{\mathrm{CT}} \times knp - k_{\mathrm{CT,diss}}\mathrm{CT} - k_{\mathrm{CT,T}}\mathrm{CT} - k_{\mathrm{CT,rec}}\mathrm{CT} \\ + k_{T,\mathrm{CT}}T + k_{S,\mathrm{CT}}S = 0, \qquad (3)$$

$$\frac{\mathrm{d}T}{\mathrm{d}t} = (1-c_S)(1-c_{\mathrm{CT}}) \times knp + k_{\mathrm{CT,T}}\mathrm{CT} - k_{T,\mathrm{CT}}T \\ - k_T T - k_{\mathrm{TTA}}T^2 + 2k_{\mathrm{SF}}S = 0, \text{ and} \qquad (4)$$

$$\frac{\mathrm{d}S}{\mathrm{d}t} = c_S(1-c_{\mathrm{CT}}) \times knp + \tfrac{1}{2}k_{\mathrm{TTA}}T^2 - k_{\mathrm{SF}}S - k_{\mathrm{rad}}S \\ - k_{\mathrm{nonRad}}S - k_{S,\mathrm{CT}}S = 0, \qquad (5)$$

where $k$ is the recombination rate of free charge carriers, $k_{\mathrm{TTA}}$ the TTA coefficient, $k_T$ the recombination rate of triplets $T$ to the

ground state, $k_{SF}$ the singlet-fission rate, and $k_{rad}$ and $k_{nonRad}$ the rates at which singlets in rubrene recombine radiatively and non-radiatively. For simplicity, we do not distinguish between radiative and non-radiative recombination of triplets. The generation and recombination or dissociation of bound CT states at the heterojunction interface are described by $k_{CT}$, $k_{T,CT}$, $k_{CT,P}$, $k_{CT,rec}$, and $k_{S,CT}$. For energetic reasons, we do not allow for generation of singlet excitons from CT states; as the energy of the triplet state in rubrene and the CT state nearly coincide, exchange between the two species was included. Note that back transfer from rubrene triplet states to CT states is penalized energetically and a small rate constant is to be expected. Often, instead of recombination rates, the lifetimes for certain processes are found in the literature, $\tau = k^{-1}$. The proportionality constant $c_S$ determines the fraction of singlet to triplet excitons generated from free charge carriers; under normal device operation $c_S = 1/4$. In a similar manner, the constant $c_{CT}$ determines the fraction of carrier recombination generated CT states vs. the fraction of directly generated singlets and triplets in rubrene. Thus, $c_{CT}$ represents the branching between the bulk singlet and triplet channels, captured in the drift-diffusion model, and parallel charge recombination channels at the interface. The luminescence intensity is proportional to $L \propto k_{rad}S$. Equations (2) to (5) form a system of non-linear equations and simulated luminescence data will be shown later in the manuscript.

It is illustrative to simplify the equations to look at limiting cases. Neglecting the generation of singlets from free charge carriers ($c_S = 0$), singlet fission, and generation of CT states from triplets, we can estimate the current density dependence of the luminescence $L(J)$. This represents the recent model by Qiao[42]. In this case, assuming low-triplet-state densities, Eq. (4) leads to a linear relationship between triplet state density and injected current $T \sim \frac{\tau_T}{q a_0} J$, and via Eq. (5) to a quadratic dependence for the luminescence intensity with respect to the injected current density $L \propto S \sim J^2$. In case of a high triplet density, $T > (\tau_T k_{TTA})^{-1}$ or very long triplet lifetimes, Eq. (4) can be simplified to $T \sim \sqrt{\frac{1}{k_{TTA} q a_0}} J$. In this case, Eq. (5) yields a linear luminescence dependence $L \propto S \sim J$. In the case of a coherent process, and neglecting other recombination types, Eq. (2) essentially must be replaced by:

$$\frac{dn}{dt} = \frac{dp}{dt} = \frac{J}{q a_0} - k_{coh} n^2 p^2 + k_{CT,diss} CT = 0, \quad (6)$$

which, neglecting singlet exciton generation from free charge carriers, leads to the same current dependencies for low-triplet-state densities $L \propto S \sim J^2$ and high-triplet-state density $L \propto S \sim J$.

Shown in Fig. 8 are the experimental $L(J)$-characteristics, $\partial \log(L)/\partial \log(J)$ and current efficiencies as function of device current within the exponential region of the $J(V)$- and $L(V)$-data, where the device is minimally affected by either series or parallel resistance. For small currents, we observe a near unity dependence of $L(J)$ vs. current through the device, while for medium-to-high currents, the devices with a thin BCP interlayer exhibit an exponent above 1. The exponent increases from 0 nm BCP with increasing BCP thickness, reaching its maximum at 4 nm BCP before decreasing for thicker films. In all cases, for high-current densities, the exponent decreases back to unity, as the device becomes more and more series-resistance limited. These observations are contrary to the simple model prediction of $n \approx 2$ at low current (triplet density) and are in contrast to the recent results by Qiao[42], who observed an exponent of 2 for small current. Our results suggest that while TTA may be observed experimentally in medium-to-high current densities, at low-

current densities, the simplified model fails to describe the device behavior.

In the following, we examine the $L(J)$-characteristics when generation of singlet excitons from free charge carriers, via minority carrier recombination, and singlet fission are not neglected. The rate constants for the triplet decay, singlet fission, and TTA in rubrene have been determined by others from time-delayed fluorescence, time-resolved photoluminescence, and transient absorption spectroscopy. Ryasnyanskiy and Ivan Biaggio measured a triplet lifetime $\tau_T$ in single rubrene crystals on the order of 100 μs and a recombination constant $k_{TTA}$ on the order of $10^{-14}$ cm$^3$/s[43]. The singlet-fission rate and radiative recombination rate in solutions and films was investigated by Wen et al.[44], Piland et al.[45], Ma et al.[46,47], Jankus et al.[48], Tao et al.[49], and others. In case of singlet fission, two characteristic lifetimes $\tau_{SF}$, with the shorter lifetime being between 2 and 6 ps, and a second lifetime, delayed due to phonon scattering, on the order of 20–50 ps were observed. Radiative recombination in solution was about 16 ns. Assuming a non-radiative lifetime of singlets on the order of the phonon-scattering time (50 ps), we can readily determine the $L(J)$-characteristics and the power law dependence of $L \propto J^n$.

Simulated $L(J)$-characteristics and $\partial \log(L)/\partial \log(J)$ as function of device current within the exponential region of the $J(V)$- and $L(V)$-data using the rate equations (2)–(5) are shown in Fig. 9. The fraction of singlet excitons generated from free charge carriers was varied between 0 and 1/4, corresponding to the above discussed extreme case and more realistic device operation. The rate constants involving the CT state are not known. However, their nominal dependence on BCP layer thickness can be proposed based on simple physics and existing literature. Insertion of the insulating spacer will lower the binding energy of the CT state[21,22] and therefore decrease $k_{CT,rec}$ (energy gap law and wavefunction effects[20,22,23,50]), slightly decrease $c_{CT}$ (lower driving energy) and slightly increase $k_{CT,diss}$[22,23]. Additionally, the spacer should decrease $k_{S,CT}$ due to changes in the wavefunction overlap between the molecular singlet and charge-separated CT exciton[22,23,50]. In Fig. 9, $k_{S,CT}$ and $k_{CT,rec}$ are assumed to obey a decaying exponential form, consistent with previous literature. The weakly varying rates are assumed constant. In Supplementary Note 5 we explore the degree at which each of the rates contribute to the simulations. Shown in Supplementary Figure 5 are simulations assuming that only $k_{S,CT}$ is changed by the introduction of the BCP interlayer; Supplementary Figure 6 shows simulations assuming that $k_{CT,rec}$ is the sole thickness-dependent rate. However, from a simulation of the short-circuit density $J_{sc}$ shown in Supplementary Figure 7, it is evident that the decrease in $J_{sc}$ with increasing interlayer thickness is due to the thickness dependence of $k_{S,CT}$ and cannot be described by the thickness dependence of $k_{CT,rec}$ alone.

The simulated data for neglectable singlet generation of free charge carriers reproduces the above-derived dependencies for the simple model: low $L \propto S \sim J^2$ and high-triplet-state density $L \propto S \sim J$. Once generation of singlet states from minority charge injection is included, the low-current density luminescence becomes linear with current through the device and reproduces our experimentally observed results of $\partial \log(L)/\partial \log(J)$. For low-current densities near the turn-on voltage, these singlets dominate the luminescence characteristics and singlets generated via TTA only correspond to a small fraction of the singlet population. Only for medium- and high-current densities is TTA of significant importance leading to enhanced light emission. Two critical observations arise from the kinetic model. The first is the need to have a direct singlet generation channel (bulk formation from minority carrier injection) to reproduce the linear behavior at low current. The second is the role of interface quenching in

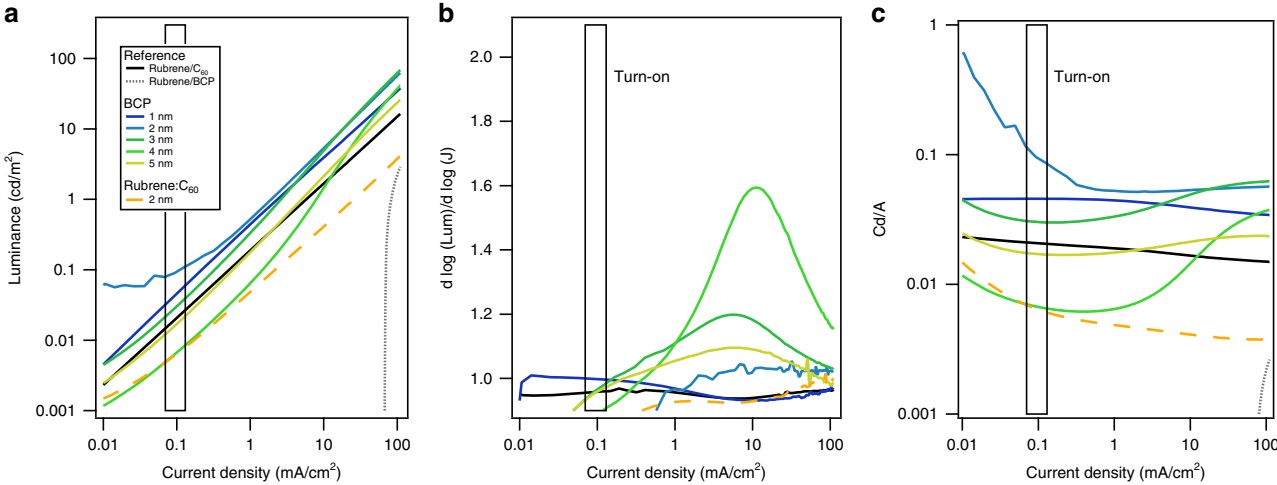

**Fig. 8** Detailed luminescence characteristics. **a** $L(J)$-characteristics of rubrene/BCP and rubrene/$C_{60}$ OLEDs as a function of interlayer thickness. **b** $\partial \log(L)/\partial \log(J)$ and (**c**) current efficiency for the same devices as shown in **a**. The beginning of the "turn-on" area is indicated. Note that at high-current densities (>100 mA/cm²), the device is series-resistance limited

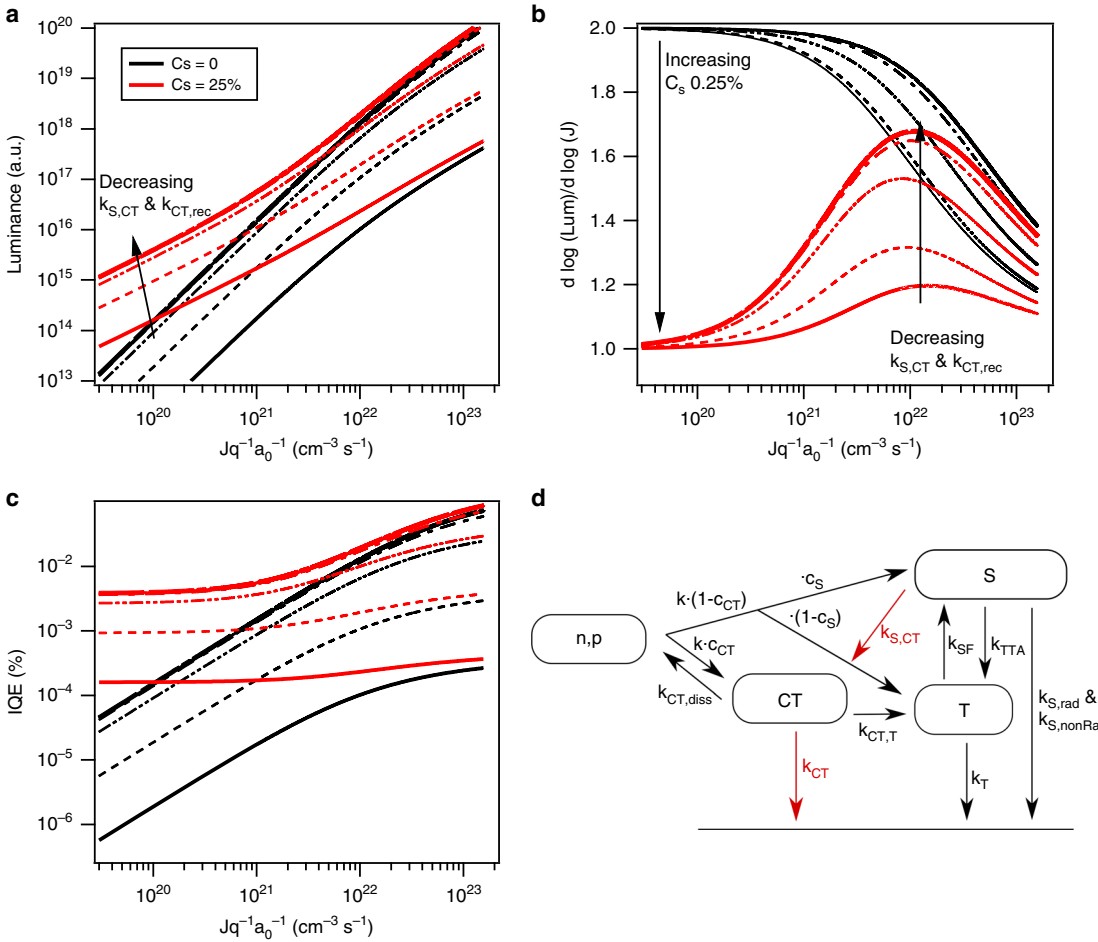

**Fig. 9** Kinetic model results. **a** Simulated $L(J)$-characteristics, (**b**) $\partial \log(L)/\partial \log(J)$, and (**c**) current efficiency of rubrene-based OLEDs. The data were calculated using Eq. (2)-(5) and literature values for $k_{TTA} = 10^{-14}$ cm³/s, $\tau_T = 100$ μs, $\tau_{SF} = 5$ ps, $\tau_{nonRad} = 50$ ps, and $\tau_{rad} = 15$ ns, $c_{CT} = 50\%$, and $k_{T,CT} = 0$. Assumed values: $\tau_{CT} = 50$ ns; $\tau_{CT,T} = 10$ ps, $\tau_{S,CT} = 10$ ps. The fraction of directly generated singlets, $C_s$, has been simulated for 0 and 25%. To simulate the introduction of a thin BCP layer, the rates $k_{CT}$ and $k_{S,CT}$ have been multiplied by an exponential term of the form exp($-d/d_0$), with $d_0 = 0.5$ nm and $d = 0...5$ nm. All other rates are assumed to be either independent (singlet fission, TTA, ...) or only weakly varying (CT dissociation, ....) with BCP layer thickness. Shown in **d** is a schematic representation of the possible recombination pathways. The exponentially varied rates are highlighted in red in the schematic summary of included processes shown in **d**

determining the luminescence from interface states. While either a decrease in $k_{S,CT}$ or $k_{CT,rec}$ can lead to increased CT states and increased TTA with increasing BCP thickness, only a decrease in $k_{S,CT}$ is simply consistent with both the increase in luminescence and decrease in OPV $J_{sc}$ for the modified devices.

We note in passing that prior reports on rubrene/$C_{60}$ LEDs have relied on MEL studies to identify higher order spin-dependent up-conversion processes as the source of the half-bandgap turn-on voltage. We find the MEL response to lack the specificity needed to assign TTA as the dominant process involved in the emission process (S1). On the other hand, our electrical analysis provides clear evidence that neither Auger or TTA processes significantly affect the device current.

## Discussion

We have probed the role of interface kinetics in rubrene/$C_{60}$ heterojunction OLEDs by selective modification of the junction. We show that the introduction of a 2 –3 nm-thick BCP interlayer leads to a threefold increase in device luminance, while maintaining the characteristic half bandgap turn-on. The detailed device studies clearly indicate that neither Auger recombination (diode ideality of 2/3) nor TTA (non-linear $L(J)$ behavior) are necessary (or consistent) with the behavior of the baseline, rubrene/$C_{60}$ device. The success of the drift-diffusion model establishes that low-voltage EL in OLEDs can simply arise from the low built-in potential of a heterojunction interface and minority carrier recombination, as is known for inorganic LEDs, in complete agreement with the generalized Planck equation. We do find that, by tuning the interfacial rates, an interface state mediated up-conversion process can be revealed. Critical to this is the understanding that heterojunctions can act to dissociate singlets, thus modification of this parasitic channel is critical to observation of interface generated singlets. This is in contrast to studies where the CT states (exciplexes) are emissive, where the role of bulk singlet quenching is avoided.

## Methods

**Device fabrication**. To systematically vary the electron-hole separation distance between rubrene and $C_{60}$, we employed the following device structure: indium-tin-oxide (145 nm)/PEDOT:PSS:Nafion (20 nm)/rubrene (40 nm)/BCP or rubrene:$C_{60}$ interlayer (0 –5 nm)/$C_{60}$ (20 nm)/BCP (10 nm)/Al (90 nm). A reference rubrene-only device was also created with the structure: ITO (145 nm)/PEDOT:PSS:Nafion (20 nm)/rubrene (40 nm)/BCP (10 nm)/Al (90 nm). Devices were deposited on commercially available patterned indium tin oxide (thin film devices (Certain commercial equipment, instruments, or materials are identified in this paper in order to specify the experimental procedure adequately. Such identification is not intended to imply recommendation or endorsement by the National Institute of Standards and Technology, nor is it intended to imply that the materials or equipment identified are necessarily the best available for the purpose.), $R_s$ = 15 ohms/sq). Substrates were cleaned via sonication in deionized water, acetone, and isopropanol, followed by 15 min of ultraviolet/ozone treatment. Next, a mixture of PEDOT:PSS and Nafion by mass (9:1) was spin-coated at 523 rad s$^{-1}$ (5000 r.p.m.) for 1 min and annealed at 140 °C for 20 min. The rubrene (Sigma-Aldrich, >99.99), BCP (Lumtec >99 %), $C_{60}$ (M.E.R. Corporation, >99.9%), and Al (R.D. Mathis, >99.99 %) were thermally evaporated using a Lesker deposition system at a base pressure of <1.2 10$^{-4}$ Pa connected to an Argon purged glovebox (<5 ppm $H_2O$, $O_2$). The active area, as defined by the overlap of ITO and metal contacts, was 4 mm$^2$.

**Device characterization**. Current–voltage, $J(V)$, characteristics were measured using a Keithley 2636 A source meter. Absolute luminance values were measured with a Photo Research PR880 photometer at drive currents of 0.2 , 1 , 10, and 20 mA, and mapped onto a relative $L(V)$-curve measured by a Si-photodetector (Thorlabs SM1PD1B). MEL was measured in air on devices encapsulated inside an Ar filled glovebox placed between the poles of an electromagnet, with the magnetic field vector perpendicular to the surface normal. The magnetic field was measured using a Hall probe GaussMeter with a maximum strength of 100 mT and was applied parallel to the device surface. The EL spectra were measured with an Ocean Optics QE65PRO spectrometer. Drive currents for MEL and EL spectra were 0.75 mA. Solar cell $J$–$V$ measurements were measured under AM1.5 conditions.

## Data availability

The data that support the findings of this study are available from the corresponding author upon reasonable request.

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

## Acknowledgements

A.J.B. acknowledges support from a National Institute of Standards and Technology National Research Council fellowship. We want to thank Kurt Pernstich and Beat Ruhstaller from the Optoelectronic Research Laboratory, ZHAW School of Engineering for help full discussions and feedback. We thank Roderick MacKenzie for helpful discussions on the use of GPVDM.

## Author contributions

A.J.B. performed device optimization and device measurement, and S.E. performed device analysis and modeling. E.G.B. provided experimental support. S.E. and A.J.B. wrote the manuscript. N.C.G. provided critical device modeling insights, L.J.R. and D.J.G. oversaw experimental design, analysis, and writing. All authors have given approval to the final version of the manuscript.

## Additional information

**Competing interests:** The authors declare no competing interests.

