## [Peer Review File · Nature Communications]

Reviewers' comments:

Reviewer #1 (Remarks to the Author):

Authors report on the role of the charge transfer (CT) state in Rubrene/C60 OLEDs with low-voltage turn-on. The work is very important in view of previous research carried about 10 years ago, into which previous researchers attributed their discovery of a low-voltage turn-on in OLEDs to an Auger fountain electroluminescence, which attribution was inspired from the physics of compound semiconductor diodes. In organic films, the Auger fountain electroluminescence process must be mediated by a CT state at the interface between the two components of the diode to reach significant cross-section.

By tuning the interface CT using BCP, authors clearly demonstrate that the process cannot be Auger-type. Additional modeling of the IV characteristics confirms that the 3-particle Auger process can be ruled out from their experiments. They then revisit previous experiments on triplet fusion into a luminescent singlet, which happens to be the only valid explanation to the low-voltage turn-on.

All the methodology and the interpretation makes a lot of sense, literature is adequately cited and the manuscript is well written in a clear and concise style.

Work is important not just to revisit previous publications, but it is necessary to clearly identify the mechanisms in order to create new principles and applications related to the handling of triplet excitons in organic semiconductors.

Reviewer #2 (Remarks to the Author):

In this manuscript Engmann et al. report on the origin of the half-bandgap electroluminescence from Rubrene/C60 heterojunction devices. These authors evaluated various device structures with an inclusion of a thin BCP interlayer or a mixed Rubrene:C60 interlayer at the Rubrene/C60 interface. Based on device characteristics from OLED and OPV with various BCP and Rubrene:C60 interlayer thicknesses, the authors conclude that CT-state is not a necessary intermediate step in the half-bandgap electroluminescence. Based on numerical modeling analysis, the authors also conclude that band-to-band bimolecular recombination, rather than the Auger-assisted upconversion nor the TTA, is the origin of the half-bandgap turn-on. I have two major concerns about this conclusion: (1) the fundamental physics on the energy upconversion from applied bias voltage x eV to emitted photon energy $\sim 2x$ eV is not clearly discussed; (2) electron-hole wave function overlap and the energy gap law of non-radiative recombination at the Rubrene:C60 interface are completely ignored in this manuscript.

Other technical comments are:

1) The authors state "insert a thin BCP interlayer or a mixed Rubrene:C60 interlayer in Rubrene/C60 interface to suppress or enhance the formation of the CT-state and recombination rate". How can the authors prove the variation of CT-state formation and recombination? Sampat et al. [J. Phys. Chem. C 2015, 119, 1286–1290] have demonstrated that increasing thickness of the interlayer (LiF) film from 0 to 1 nm significantly suppresses CT excitons recombination while CT excitons generation remains practically unaltered.

2) On page 6, paragraph 2, the authors state, "This may be a trap assisted hopping of electrons across the interlayer similar to that observed at the C60/BCP/metal electrode, direct tunneling, or percolating across a non-continuous layer." How can the authors prove this? Nakanotani et al. [Sci. Adv. 2016, 2 : e1501470] and He et al. [J. Phys. Chem. C 2016, 120: 21325–21329] have found electrons in the LUMO of acceptor can form CT excitons with holes in the HOMO of donor through a long range electron-hole coupling across an interlayer.

3) Based on the idea that the CT-state decrease with the thickness of BCP layer and the experimental data that the luminance intensity from Rubrene increase with thickness of BCP interlayer, the authors assert "CT-state is not a necessary and relevant intermediate step in the formation of the high-energy singlet emission, rather, the CT-state acts as competing process."

This assertion is cursory because the BCP interlayer may suppress the CT-state recombination leading to increased luminance intensity in Rubrene through TTA process.

4) Introduction of a mixed Rubrene:C60 interlayer leads to a precipitous decrease in luminescence with increasing thickness. The author assert "an increased CT-state density suppresses luminescence from Rubrene." This assertion is also cursory because of that the Rubrene:C60 interlayer will reduce the interfacial electron concentration at Rubrene/C60 heterojunction. He et al. [Adv. Mater. 2016, 28, 649–654] have shown that Auger electronic process at organic heterojunction is dominated by the accumulated electron concentration at the donor/acceptor interface.

5) The OPV characteristics for the various BCP and Rubrene:C60 interlayer thicknesses are also used to confirm the author assertions. The author state "with increasing BCP interlayer thickness, a monotonic decrease in short circuit current density (J_{sc}) is observed. This agrees well with our expectations that the formation of the CT-state is suppressed by the BCP interlayer as splitting of the rubrene exciton at the rubrene/C60 interface is necessary for solar cell operation." However, a monotonic J_{sc} decrease in BCP interlayer devices cannot explain decrease in CT-state density. This is because the energy barrier between Rubrene and BCP will reflect singlet excitons from the interface and thus reducing photo-carrier generation efficiency.

Other minor comments:

1) Page 3, Line 74, the work function of PEDOT:PSS:Nafion is usually in the range of 5.5 to 5.9 eV. Please cite the correct values.

2) Page 6, Figure 3, please state the driving condition for the EL spectra being recorded. Are they measured at the same voltage or same current level?

3) Page 11, Table 1, there is a typo for the nLum of the Rubrene/BCP diode.

4) Page 18, line 318, the equation for R seems wrong.

5) Typo on page 18, line 326, "HOMO / LUMO energies or rubrene and C60".

The Role of Higher Order Effects in Rubrene/C₆₀ Organic Light-Emitting Diodes with Low-Voltage Turn-On

RESPONSE TO REVIEWERS

We appreciate the informative feedback and constructive criticism from the reviewers. We have addressed each comment/question by the reviewers as detailed below. Original referee comments are listed in **black**, our responses are in **blue**, and revisions in the manuscript are highlighted in **yellow**.

Reviewer #2 (Remarks to the Author):

Based on numerical modeling analysis, the authors also conclude that band-to-band bimolecular recombination, rather than the Auger-assisted up conversion nor the TTA, is the origin of the half-bandgap turn-on. I have two major concerns about this conclusion: (1) the fundamental physics on the energy up-conversion from applied bias voltage \times eV to emitted photon energy $\sim 2x$ eV is not clearly discussed; (2) electron-hole wave function overlap and the energy gap law of non-radiative recombination at the Rubrene:C60 interface are completely ignored in this manuscript.

As was clearly discussed in the first paragraph of the conclusion, from a thermodynamic perspective, the energy of emitted photons is not constrained by the applied voltage (ref 36 in the original manuscript, ref 28 in the revised manuscript). Also, as discussed in the conclusion, high photon energy emission arises simply from the minority carrier recombination current of a heterojunction pn-diode and *does not* require any exotic up-conversion physics (Refs 37-41 of the original manuscript and 44-48 of the revised manuscript). This is detailed numerically in the section titled 'A drift-diffusion model for rubrene/C₆₀ without the need for TTA' that has been moved up to pg 11 in the restructured manuscript. A more elaborate presentation of these points in the manuscript is precluded by length constraints, but we present them below.

From a thermodynamic perspective, light emission from semiconductors is governed by the generalized Planck equation derived by Wurfel in Ref. [28]:

$$I(E) = A(E) \frac{2\pi E^2}{h^3 c^2} \left[\frac{1}{\exp[(E - \mu_{eh})/k_b T] - 1} \right],$$

where I is the spectral flux of photons with energy E , μ_{eh} is the local difference between electron and hole chemical potentials, and $A(E)$ is the absorbance spectrum of the semiconductor. In a pn junction LED with negligible series resistance, μ_{eh} is simply equal to the applied voltage and thus $I \propto \exp(qV/k_b T)$. That is, there is no fundamental voltage threshold for light emission – the 'turn-on' voltage depends only on the sensitivity of the photon detection apparatus. In fact, the only light emission situation where the voltage is required to exceed the bandgap is to achieve stimulated emission (i.e. a population inversion).

The same conclusion follows from a device physics perspective if one considers the ideal diode equation for a pn junction. In this case, the current density $J \propto \exp(qV/k_bT)$ arises from minority carrier diffusion, which implies that the minority carriers are recombining with majority carriers. In a direct bandgap semiconductor, these recombination events take place radiatively (in proportion to the luminescence quantum yield) and thus the emitted light intensity carries the same, thresholdless exponential voltage relationship described above.

In regard to major concern #2, while the energy gap law and degree of electron-hole overlap at the Rubrene:C₆₀ interface certainly impact the lifetime and spatial extent of the CT state, neither has any direct bearing on the band-to-band recombination process that we conclude underlies Rubrene electroluminescence in our devices per the discussion above. We have, however, introduced the concepts and relevant literature in the discussion of the kinetic model (pg 18, highlighted)

Other technical comments are:

The following comments of the reviewer clearly point out that the role of CT states in OLED and OPV devices arises from a complex interplay between CT formation, recombination, and dissociation rates and our language in the early discussion of the effect of BCP or mixed layers was imprecise. We have restructured the manuscript to delay the discussion of the role of CT states to the discussion of the complete kinetic model where the reviewer's insights are addressed and have introduced qualifying text where necessary.

- 1) The authors state "insert a thin BCP interlayer or a mixed Rubrene:C60 interlayer in Rubrene/C60 interface to suppress or enhance the formation of the CT-state and recombination rate". How can the authors prove the variation of CT-state formation and recombination? Sampat et al. [J. Phys. Chem. C 2015, 119, 1286–1290] have demonstrated that increasing thickness of the interlayer (LiF) film from 0 to 1 nm significantly suppresses CT excitons recombination while CT excitons generation remains practically unaltered.

A complete, independent determination of all the rate processes that involve CT states is beyond the scope of this manuscript. We have modified the sentence in question (pg 2 highlighted) to make clear that we are highlighting the effect of the BCP layer on the Singlet to CT channel, not the free charge to CT channel. Additionally, we adopt the phrase 'interface pathways' (pg 3 highlighted) to reflect the total effect of the CT state kinetics, avoiding approximation to individual rates until the kinetic model is introduced. The kinetic model we introduce is more complete than preceding models in the literature (Liu, ref 24, Giebink, ref 27 & 28 and Qiao 43) and allows us to parametrically explore the role of CT-state formation and recombination kinetics in our device. We thank the reviewer for pointing out that CT-state recombination was not in our original model. We have included it (pg 13 highlighted) and our conclusions are now even more robust

We point out that Sampat, who studied the tetracene|LiF|C60 OPV system first demonstrated by Campbell, optically pumped the interfaces and demonstrated that 2 nm of LiF effectively *eliminated* CT state formation from bulk singlets (the slow, diffusion limited channel). The prompt CT state formation that remained is due to *direct* excitation of the interface by the

intense pump pulse, a CT state creation channel that is not present in our OLED devices (ignoring self-absorption).

The consensus of the literature discussed by reviewer #2 is that wide band gap spacer layers should:

Lower the binding energy of the CT state (Campbell, Nakanotani) and therefore decrease the non-radiative recombination rate (energy gap law & wavefunction effects Campbell (2012) ref 23, Liu (2013) ref 24, He (2016) ref 21, Sampat (2015) ref 51), slightly decrease the formation rate from free carriers (lower driving energy) and slightly increase the dissociation rate to free carriers (Campbell, Liu).

Additionally, the spacer should decrease the singlet to CT formation rate due to changes in the wavefunction overlap between the molecular singlet and charge separated CT-exciton (Campbell, Liu, Sampat)

We now explicitly discuss these dependences in the manuscript on pg 18.

- 2) On page 6, paragraph 2, the authors state, "This may be a trap assisted hopping of electrons across the interlayer similar to that observed at the C60/BCP/metal electrode, direct tunneling, or percolating across a non-continuous layer." How can the authors prove this? Nakanotani et al. [Sci. Adv. 2016, 2 : e1501470] and He et al. [J. Phys. Chem. C 2016, 120: 21325–21329] have found electrons in the LUMO of acceptor can form CT excitons with holes in the HOMO of donor through a long range electron-hole coupling across an interlayer.

The key observation on pg 6 is that the *device current* was independent of BCP layer thickness. Barring pathological correlations between the multiple rate processes at the interface, this indicates the interface recombination is unlikely the dominant recombination process in the device. Specifically, the anticipated exponential decrease in CT state non-radiative recombination with BCP layer thickness, observed by both Campbell and Liu and He (see above) is not observed. We have inserted text on page 6 (highlighted) to make this discussion clear.

We note that, while Nakanotani, et al. beautifully demonstrate that radiative CT states can be supported across thick BCP layers (and is now cited on pg 8, highlighted), they do not report the consequences of the layer thickness on total device current, the point of the discussion on pg. 6. They only report EQE vs Luminance.

- 3) Based on the idea that the CT-state decrease with the thickness of BCP layer and the experimental data that the luminance intensity from Rubrene increase with thickness of BCP interlayer, the authors assert "CT-state is not a necessary and relevant intermediate step in the formation of the high-energy singlet emission, rather, the CT-state acts as competing process." This assertion is cursory because the BCP interlayer may suppress the CT-state recombination leading to increased luminance intensity in Rubrene through TTA process.

We agree that, at this stage of the manuscript, the assertion is cursory, although it is held-up by the detailed kinetic model. CT-state recombination cannot be rate limiting in the baseline device, due to the monotonic decrease in current with BCP layer thickness under OVP operation.

The text on page 7-8 (highlighted) has been changed to reflect the complexity of CT state kinetics and the properly supported conclusion.

- 4) Introduction of a mixed Rubrene:C60 interlayer leads to a precipitous decrease in luminescence with increasing thickness. The author assert “an increased CT-state density suppresses luminescence from Rubrene.” This assertion is also cursory because of that the Rubrene:C60 interlayer will reduce the interfacial electron concentration at Rubrene/C60 heterojunction. He et al. [Adv. Mater. 2016, 28, 649–654] have shown that Auger electronic process at organic heterojunction is dominated by the accumulated electron concentration at the donor/acceptor interface.

The assertion has been removed from the manuscript, as it is not necessary to the conclusions that higher-order processes at the interface are not necessary to describe the baseline device operation, but maybe revealed by judicious modification of the relevant interface rates.

- 5) The OPV characteristics for the various BCP and Rubrene:C60 interlayer thicknesses are also used to confirm the author assertions. The author state “with increasing BCP interlayer thickness, a monotonic decrease in short circuit current density (J_{sc}) is observed. This agrees well with our expectations that the formation of the CT-state is suppressed by the BCP interlayer as splitting of the rubrene exciton at the rubrene/C60 interface is necessary for solar cell operation.” However, a monotonic J_{sc} decrease in BCP interlayer devices cannot explain decrease in CT-state density. This is because the energy barrier between Rubrene and BCP will reflect singlet excitons from the interface and thus reducing photo-carrier generation efficiency.

We are confused by the reviewer’s comment for exciton reflection is simply complete elimination of the possibility of exciton dissociation. However, the text was removed and more precise language describing the anticipated dependence of singlet dissociation on BCP layer thickness is presented in the discussion of the detailed kinetic model. For thin (>1 nm but \leq 5 nm) BCP layers, our results agree with Campbell et al. who have found that exciton dissociation can be described as tunneling of either hole or electron across the interface, thus leading to an exponential decrease in current density for a solar cell configuration.

Other minor comments:

- 1) Page 3, Line 74, the work function of PEDOT:PSS:Nafion is usually in the range of 5.5 to 5.9 eV. Please cite the correct values.

We thank the reviewer for pointing this out. We cite now a range of work functions based on references 11-15.

- 2) Page 6, Figure 3, please state the driving condition for the EL spectra being recorded. Are they measured at the same voltage or same current level?
- 3) Page 11, Table 1, there is a typo for the nLum of the Rubrene/BCP diode.
- 4) Page 18, line 318, the equation for R seems wrong.
- 5) Typo on page 18, line 326, "HOMO / LUMO energies or rubrene and C60".

We have made the suggested changes (2-5) and are grateful for the careful reading of the manuscript.

REVIEWERS' COMMENTS:

Reviewer #2 (Remarks to the Author):

The authors have done an excellent job in revising the manuscript. Most technical problems/concerns have been fixed. I, however, remain skeptical on the methodology of using thermodynamic theory to model the extremely high (~ 1 eV) photon energy above the applied bias-the Boltzmann tail for "hot" charge carriers typically won't exceed 100 meV. This referee still believe Auger-type trion recombination is behind the working of the photon energy upconversion. As this is a very important problem for organic electronics, a contrarian view is certainly a very important aspect for maintain a healthy research ecosystem and thus I recommend this manuscript for publication in Nature Communications.